

# Window films increase avoidance of collisions by birds but only when applied to external compared with internal surfaces of windows

John P. Swaddle, Blythe Brewster, Maddie Schuyler and Anjie Su

Institute for Integrative Conservation, William & Mary, Williamsburg, VA,
United States of America

Corresponding author
John P. Swaddle, jpswad@wm.edu

## ABSTRACT

Window collisions are one of the largest human-caused causes of avian mortality in built environments and, therefore, cause population declines that can be a significant conservation issue. Applications of visibly noticeable films, patterns, and decals on the external surfaces of windows have been associated with reductions in both window collisions and avian mortality. It is often logistically difficult and economically prohibitive to apply these films and decals to external surfaces, especially if the windows are above the first floor of a building. Therefore, there is interest and incentive to apply the products to internal surfaces that are much easier to reach and maintain. However, there is debate as to whether application to the internal surface of windows renders any collision-reduction benefit, as the patterns on the films and decals may not be sufficiently visible to birds. To address this knowledge gap, we performed the first experimental study to compare the effectiveness of two distinct window films when applied to either the internal or external surface of double-glazed windows. Specifically, we assessed whether Haverkamp and BirdShades window film products were effective in promoting the avoidance of window collisions (and by inference, a reduction of collisions) by zebra finches through controlled aviary flight trials employing a repeated-measures design that allowed us to isolate the effect of the window treatments on avoidance flight behaviors. We chose these two products because they engage with different wavelengths of light (and by inference, colors) visible to many songbirds: the BirdShades film is visible in the ultraviolet (shorter wavelength) range, while the Haverkamp film includes signals in the orange (longer wavelength) range. We found consistent evidence that, when applied to the external surface of windows, the BirdShades product resulted in reduced likelihood of collision and there was marginal evidence of this effect with the Haverkamp film. Specifically, in our collision avoidance trials, BirdShades increased window avoidance by 47% and the Haverkamp increased avoidance by 39%. However, neither product was effective when the films were applied to the internal surface of windows. Hence, it is imperative that installers apply these products to exterior surfaces of windows to render their protective benefits and reduce the risk of daytime window collision.

## INTRODUCTION

Bird collisions with windows kill more than a billion birds per year, creating substantial conservation and socioeconomic problems in many built areas (*Klem, 2014*; *Loss et al., 2014*; *Loss, Will & Marra, 2015*; *Ocampo-Peñuela et al., 2016*; *Schneider et al., 2018*). Therefore, there is societal need to create solutions that decrease these collisions and promote avoidance of windows (*Hager et al., 2013*; *Klem & Saenger, 2013*).

One way to reduce the probability of window collision is to treat the external surface of glass with a film or decals that should increase the visual conspicuousness of windows to birds and are more easily differentiated from suitable habitat (*Klem & Saenger, 2013*; *Rössler, Nemeth & Bruckner, 2015*; *Sheppard, 2019*). It is believed that the reflection from windows could be misperceived as habitat that the bird could fly to safely and that markings and decals sufficiently disrupt that reflection as it appears to birds (*Klem, 2009*). To these ends, researchers have conducted in-field trials of window film products to examine whether the products decrease deaths and collisions of birds who are baited into the area where the windows are placed (*Klem et al., 2004*; *Klem & Saenger, 2013*). Window film products have also been tested in forced, binary choice trials in flight tunnels, where birds have the choice to collide with (though a fine net prevents actual collision) a treated piece of glass compared with an untreated pane of glass (*Rössler, Nemeth & Bruckner, 2015*; *Sheppard, 2019*). In our experience of discussing window collisions with property owners and people who influence building appearance and design, both of these techniques seem to be accepted by legislators, architect groups, and the public as indicating which products will reduce bird-window collisions.

Though published reports investigating the effects of window treatments on bird-window interactions have placed the window treatment on the external surface of glass, anecdotal reports from birding groups and property owners suggests that many end-users ultimately place the window treatment on the internal (*i.e.,* inside the building) side of the glass. Interior placement of films, markings, and decals may often be logistically simpler and cheaper than fixing the same treatments to the external surface of a windows. This is especially the case with windows above ground level where end users might have to climb on tall ladders, erect scaffolding, or hire lifting machinery to apply the film to an exterior window surface. Therefore, we investigated window avoidance behaviors when the same film product was placed on the internal *versus* the external surface of windows, in controlled flight trials. To our knowledge, there has yet to be a direct comparison of the effectiveness of deterrents when applied to internal or external surfaces of windows.

A further limitation of current bird-window collision research is that few studies have reported the effects of window treatments on the avoidance of collisions (*Swaddle et al., 2020*). In-field studies of glass hung near bird feeders have often counted bird carcasses or markings left by birds on the glass (*Klem et al., 2004*). Flight tunnel tests do not give their test subjects the option to avoid a window collision—the birds have to collide with either a treated or a control pane of glass (*Rössler, Nemeth & Bruckner, 2015*; *Sheppard, 2019*). There is value in these collisions-occurrence tests, yet understanding whether a window treatment promotes avoidance of collision is also fundamental to understanding how and

why such a product would ultimately reduce avian collisions and mortality. Birds will continue to interact with glass and promoting avoidance is a major mechanism to reduce the risk of window collision. To help address this gap in methodology and knowledge, we designed a multi-experiment study to document window avoidance flight behaviors of zebra finches (*Taeniopygia guttata*), a small songbird, when presented with two window film treatments—those produced by BirdShades (an ultraviolet wavelength disrupting stripe pattern) and Haverkamp (orange and black diamonds in two parallel stripes).

Prior testing supports that the BirdShades product will alter bird-window interactions, leading to window avoidance and behaviors consistent with lower collision risk (*Swaddle et al., 2020*). We are not aware of published reports that test the Haverkamp window film but it appears somewhat visually similar (at least to humans eyes) to the Eckelt 4Bird V3066 product, which has support from forced, binary choice testing at the Honehau-Ringelsdorf Biological Station. The Haverkamp product appears as vertical stripes of a black and orange repeating diamond pattern (the aforementioned Eckelt product has black and orange circles in a similar arrangement). The BirdShades film contains vertical stripes of ultraviolet-disrupting patterns, to appear as a striped pattern to birds that can see in short, UV wavelengths. Zebra finches and songbirds can see in this part of the light spectrum (*Bennett & Cuthill, 1994*; *Hunt et al., 1998*; *Goldsmith & Butler, 2005*; *Hart & Hunt, 2007*; *Werner et al., 2012*; *Casalía et al., 2021*; *Olsson et al., 2021*) but humans cannot—the BirdShades film appears transparent to human eyes.

We chose to study these two films as they influence different parts of the avian-visible light spectrum. The Haverkamp film reflects in the middle to upper parts of the visible spectrum, as indicated by the orange color in its patterning. The BirdShades film influences short wavelengths of light that are visible to zebra finches but not visible to humans. It is possible that different wavelengths of light can penetrate and/or reflect from glass surfaces and that many commercially available glass filters out the majority of ultraviolet wavelengths. Hence, we predicted that the BirdShades (UV) film would be less effective when applied to the internal surface of a window compared with the external surface, whereas the Haverkamp film would have a smaller reduction in efficacy comparing internal *versus* external application to a window.

In this study, we placed BirdShades and Haverkamp films on the external and, separately, the internal surface of double-glazed replacement windows and quantified the window-avoidance flight behaviors of zebra finches in controlled flight trials in an open-air aviary. Through video analysis we quantified collision avoidance for all treatment conditions compared with interactions with untreated control windows. In addition, we performed binary choice trials in which one window was treated and the other was an untreated control. Such trials are somewhat similar to the forced, binary choice trials performed in flight tunnels (*Rössler, Nemeth & Bruckner, 2015*; *Sheppard, 2019*). We predicted that both window film products would promote avoidance of collisions, but only when the films are placed on the external surface of the windows. When the films are placed on the internal surface, we predicted that the Haverkamp film would outperform the BirdShades film.

## MATERIALS & METHODS

### Experimental subjects and general housing

We performed flight trials with 72 domesticated zebra finches in an outdoor flight aviary in Williamsburg, VA, USA (*Swaddle et al., 2020*). The zebra finches were kept in an outdoor aviary ($3 \times 3 \times 2.5$ m) separate to the experimental arena (described below) and had access to *ad libitum* Volkman science seed mix, drinking water, bathing water, and perches. We selected the experimental birds from a larger stock population we have maintained for 20 years with the condition that all experimental birds could fly well.

### Flight aviary and window treatments

The flight aviary consisted of a long, darkened release tunnel ($3 \times 1.2 \times 1.2$ m) that opened into a larger open-air, day-lit collision aviary ($8 \times 2.5 \times 2.5$ m), where two windows (Pella 250 Vinyl glass double-glazed replacement windows) were placed (Fig. 1). Birds experienced natural daylight in the collision aviary as the aviary was constructed with a fine mesh that let through daylight. Hence, the external surface of the windows experienced natural daylight during all trials. We conducted all trials between 0900 and 1130 in two time frames, November to early December 2020 and late September to October 2021. The two windows were placed into a wooden framed structure that was painted with dark paint to resemble the side of a building. Hence, to the birds it appeared that they were flying toward a building structure with two windows side-by-side. Behind each of the two framed windows, we constructed a lighting box so that the internal surfaces of each window were illuminated with artificial lighting (TaoTronics 12 W LED lamps on highest brightness setting) that were representative of residential or commercial buildings (*Emerson et al., 2022*). In this way, we could ensure there was natural daylight on the external surface of windows and realistic artificial lighting on the interior surface.

We placed the windows side-by-side within the wall structure, which extended from floor to ceiling of the flight aviary. To allow for avoidance of collision, we arranged the wall structure so that there was a 0.5 m gap on both the left and right that the birds could fly toward. A fine mist net, placed 1 m in front of the windows, prevented actual collisions (cf. *Swaddle et al., 2020*). The windows were mounted in the frame structure so that they tilted back by approximately 15° from vertical so that the birds were likely to see a reflection of the sky as they flew toward the windows.

The 72 birds were randomly assigned to four experiments ($N = 18$ in each experiment). The four experiments differed in which window film was applied to the windows and whether the film was fixed to the exterior or interior surface of the glass in the windows, to give the following experiments: (1) BirdShades film fixed to the exterior surface of glass; (2) BirdShades fixed to the interior surface of glass; (3) Haverkamp film fixed to the exterior surface of glass; and (4) Haverkamp film fixed to the interior surface of the glass. Experimenters were aware of assignments.

Within each of the four experiments, each of the 18 birds was exposed to three treatment conditions, in a balanced order so that the series of presentations and repeated exposure to the flight tunnel did not bias responses by birds. To account for among-bird variations in their flight behaviors, we applied a repeated-measures experimental design. This also

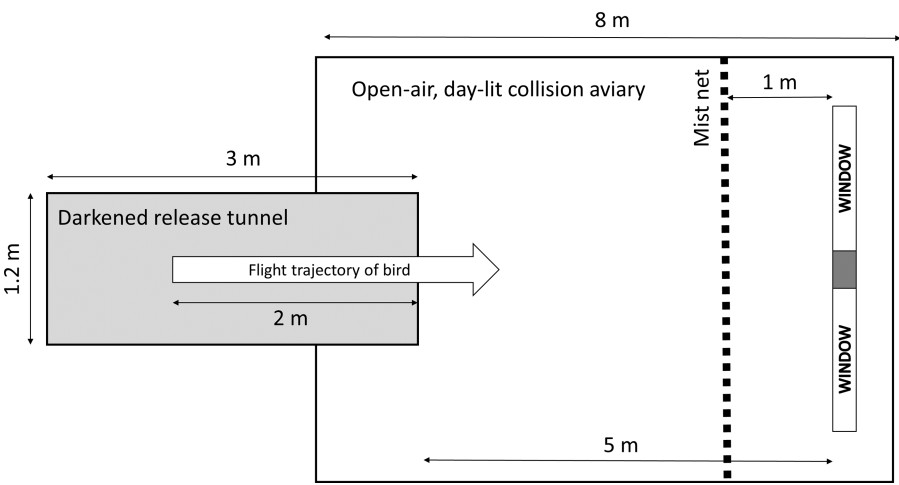

**Figure 1 Schematic of the flight tunnel.** A bird was adjudged to collide with a window if its extrapolated flight trajectory was in line with the windows and frame. As the camera was placed centrally at the end of the darkened release tunnel and birds flew in smoothly curving or straight paths, we could extrapolate flight paths relative to collision risk.

allowed us to attain greater statistical power at moderate sample sizes. As we were concerned that flight behaviors might change with repeated exposure to the flight aviary and window structure, we ensured there was at least 24 h between every experimental trial for any bird. Further, when not in the experimental trials the birds flew freely in their home aviary, which resembled the day-lit collision aviary and encouraged birds to fly actively when exposed to the treatments. The three treatment conditions were as follow:

(a) Binary choice collision trials. In these trials, one of the windows was a non-treated control while the other was treated with a window film. The control and treatment windows were equally assigned to left and right positions in the flight aviary so there was no systematic side bias. If the bird does not entirely avoid the window structure, this trial creates a forced choice situation for the birds that somewhat mimics traditional flight tunnel testing paradigms. We analyzed situations where birds collided with one of these windows to render data close to the industry-standard forced collision protocols. However, we recognize this is an unrealistic situation for a bird and conducted additional trials where our metric of interest was avoidance of collisions (*i.e.,* trials b and c, below).

(b) Control avoidance trials. We presented two non-treated windows to birds in these trials. This simulated a situation where a bird might interact with a building whose windows have not been treated with a collision mitigation film.

(c) Treatment avoidance trials. We presented two identically treated windows (according to experiments 1, 2, 3, and 4) in these trials. These trials simulated a condition where the windows of a building have been treated with a collision mitigation film. Here, the metric of interest was whether the birds avoided both windows due to the presence of the window film.

Once applied, using instructions supplied by the manufacturer, all windows treated with films were stored under heavy blankets to ensure that they remained in dark conditions when not in use, so as not to degrade or damage the film coatings. We applied the films to Pella 250 Vinyl glass double-glazed replacement windows, which is a window commonly found on residential and commercial buildings in our area.

## Flight trials and metrics of collision risk

Once the windows were placed in the frames, a flight trial commenced by releasing a single bird from the hand approximately 2 m down the dark release corridor, along with the experimenter emitting a loud startle sound. In most cases, each experimental bird flew directly toward the windows in the open collision aviary and away from the experimenter. For a bird to reach the mist net it had to fly for approximately 6 m, as the mist net was 4 m from the end of the darkened release corridor. The windows were a further 1 m behind the mist net (Fig. 1).

We recorded all flight trials on a GoPro camera (HERO8 Black at 30 frames per second; San Mateo, CA, USA) placed in the dark release tunnel and pointing toward the windows so that we could assess the probability of collision with the windows. We defined a collision as a situation when the bird flew on a trajectory that would have caused a collision if they had not hit the mist net. We defined avoidance of a collision as a situation where the bird deviated sufficiently to the right or left so that there were no longer on a trajectory to hit the windows, or if they stopped their flight at least 1 m short of the mist net and did not reach the windows (cf. *Swaddle et al., 2020*). This was done blind to treatment.

In the binary choice collision trials (condition a), we examined collisions with the treatment compared with the control window and, in addition, noted situations where birds avoided both windows. We ignored those latter situations where birds avoided both windows as we wanted to match our methodology to previous studies that have forced collisions with a window. In the control and treatment avoidance trials, we calculated the within-individual change in avoidance of the windows comparing flights when the windows were treated (condition c) to when the windows were unaltered controls (condition b). Specifically, we subtracted treatment outcomes from control outcomes for the same bird. A positive number indicates that birds were more likely to avoid windows in the treatment condition.

The procedures reported here were approved by William & Mary's Institutional Animal Care and Use Committee (IACUC-2019-09-22-13861-jpswad). The protocol was not registered elsewhere. All animals were retained after completion of the study. No animals were euthanized.

## Statistical analyses

When presented with both types of window in a binary choice trial, we used a Wilcoxon matched-pairs signed-ranks test to test whether birds were more likely to collide with a control or treatment window. For these tests, when birds flew in line with the windows, we scored collisions with a window (control or treatment) as 1 and avoidance as 0. When birds flew around the entire structure (*i.e.,* no collision was possible given the bird's flight

trajectory) that contained the windows, we scored the interaction as 0.5. As a Wilcoxon test is non-parametric and analyses rank data, the actual numerical values that we chose were not important as long as the ranks were maintained. Our approach allowed us to examine the within-individual preference for colliding with (or avoiding) control *versus* treatment windows while not skewing the outcomes of our analysis when a bird avoided both windows (as we assigned a mid-rank value of 0.5 when birds avoided both windows) (cf. *Swaddle et al., 2020*). We examined each experiment separately.

We used a Wilcoxon matched-pairs signed-ranks test to examine the within-individual change in probability of window avoidance for birds in the control (condition b) compared to the treatment (condition c) situations where we scored avoidance of the windows as 0 and collisions with windows as 1. We analyzed each experiment separately.

We also compared avoidance behaviors across experiments with a Mann Whitney U test. Specifically, we explored the effects of exterior *versus* interior application of window films by comparing the within-individual change in avoidance metric for Haverkamp and BirdShades experiments when the films were fixed to the exterior (experiments 1 and 3 combined) compared to interior surface of the glass (experiments 2 and 4 combined).

For all the analyses above, we explored mixed model options but could not get our models to converge. Hence, we employed nonparametric tests that made few assumptions about data and error distributions.

We performed all analyses using R Studio 2022.07.1 (*R Core Team, 2020*), interpreting two-tailed tests of probability (see Data S1 and S2 for data and code, respectively).

## RESULTS

### Binary choice collision trials

In binary choice collision trials with BirdShades on the external surface of the windows (experiment 1), 10 of the 18 birds were observed to collide with windows (nine control, one treatment). Overall, birds were nine times more likely to collide with unaltered controls than the BirdShades-treated window ($V = 49.5$, $df = 17$, $P = 0.013$; Fig. 2A).

When the Haverkamp film was applied to the external surface of windows (experiment 3), 12 of the 17 birds (one bird failed to complete one flight and was excluded) collided with a window (nine control, three treatment). Of those binary choice trial collisions, birds were 3-times more likely to collide with unaltered control window compared with the Haverkamp-treated window though this difference in instances of collision was not statistically supported ($V = 58.5$, $df = 16$, $P = 0.091$; Fig. 2B).

In the binary choice collision trials when the products were applied to the internal surface, there was no compelling evidence that installing either the BirdShades (experiment 2) or the Haverkamp film (experiment 4) influenced the birds' choice of collision. Specifically, when BirdShades was applied to the internal surface of windows 11 of the 18 birds collided with a window (seven control, four treatment). Of those 11 collisions there was a fairly equal number with the unaltered control and the BirdShades-treated window ($V = 42$, $df = 17$, $P = 0.392$; Fig. 2C). When the Haverkamp product was applied to the internal surface of windows, 10 of the 18 birds collided with windows (five control, five treatment).

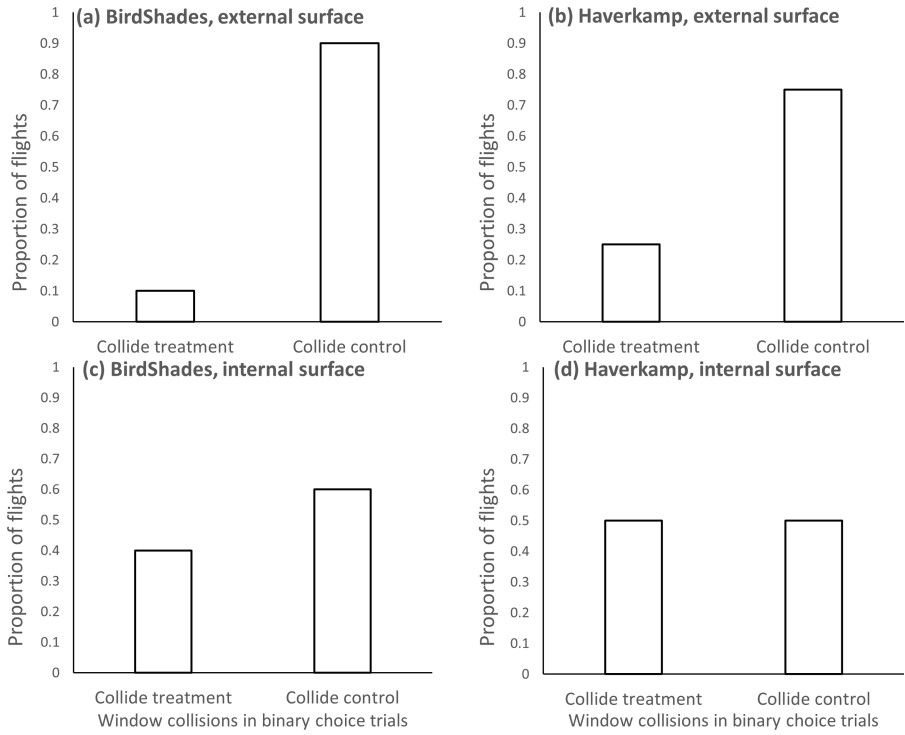

**Figure 2** **Outcomes of binary choice collision trials.** Proportion of binary choice collision trials where birds collided with either a treatment window (BirdShades shown on the left in panels (A and C) Haverkamp shown on the right in panels (B and D) or with unaltered control windows, when the treatment is applied to the external surface (shown in the upper graphs in panels A and B or the internal surface (shown in the lower graphs in panels C and D) of the windows.

Of these 10 collisions, no more occurred against unaltered controls as compared with Haverkamp-treated windows ($V = 27.5$, $df = 17$, $P = 0.999$; Fig. 2D).

### Avoidance trials

Application of the BirdShades film to the external surface of windows resulted in greater avoidance of window collisions (experiment 1: $V = 36$, $df = 16$, $P = 0.006$). One bird failed to complete one flight. The metric on the $y$-axis of Fig. 3, the mean within-individual change in avoidance (control minus treatment), is an indicator of effect size and accounts for individual changes in flight behavior. According to this metric, we observed approximately 47% more avoidance of windows when BirdShades was applied to exterior surface of the windows compared when the same birds flew toward untreated control windows.

There is marginal evidence that avoidance also occurred when the Haverkamp film was applied to the external surface of windows (experiment 3: $V = 88$, $df = 17$, $P = 0.076$). Our metric of effect size indicated that we recorded approximately 39% more window avoidance when the Haverkamp film was applied to the exterior surface of windows compared to when the same birds flew toward untreated control windows (Fig. 3).

However, there is no statistically supported evidence that application of either BirdShades (experiment 2: $V = 33$, $df = 16$, $P = 0.565$; one bird failed to complete one flight.) or

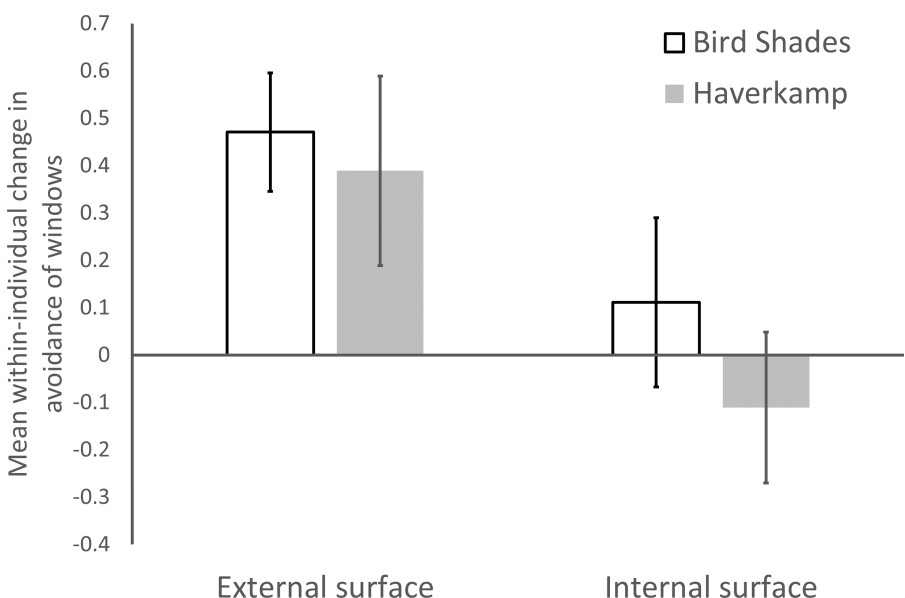

**Figure 3 Outcomes of collision avoidance trials.** Mean (±SEM) within-individual change in avoidance of collisions with windows. A positive score indicates that birds were more likely to avoid collision when a treatment (BirdShades in white, Haverkamp in gray) was applied to the windows.

Haverkamp (experiment 4: Wilcoxon $V = 13.5$, $df = 17$, $P = 0.530$) products to the internal surface of windows increased window avoidance behaviors. Note that error bars of effect size largely overlapped with zero in both of these cases (Fig. 3).

Combining the within-individual change in window avoidance data from experiments 1 and 3 (exterior fixed window film) and comparing that to experiments 2 and 4 (interior fixed window film), there was evidence that birds were more likely to avoid windows when the films were mounted on the exterior glass surface of the windows compared with when it was mounted to the interior surface ($W = 823$, $df = 34$, $P = 0.008$).

## DISCUSSION

When either the BirdShades or Haverkamp films were applied to the exterior surface of double-glazed windows we noted an increase of avoidance behaviors and decrease of collisions. The BirdShades film increased avoidance behavior by 47%, while the Haverkamp film increased avoidance by almost 40% but this latter effect was not statistically supported. Increasing avoidance behaviors is one of the ultimate goals of window collision mitigation studies but researchers and industry rarely report window avoidances and focus on events that lead to window collisions. If the window avoidance rates that we report are translated to decreases in avian mortality due to window collisions, both of these products have the promise of contributing to bird conservation outcomes.

Binary choice collision trials supported these patterns. When given a choice to collide with a treated window compared with an untreated control, birds were 9-times more likely to collide with a control window when presented with a BirdShades-treated window and

3-times more likely to hit a control window when presented with a Haverkamp-treated window. As with the avoidance trials, the numerical difference we observed with Haverkamp trials was not statistically supported. The American Bird Conservancy (ABC) uses these types of trials and data to rate window film products according to a "Threat Factor" (https://abcbirds.org/glass-collisions/threat-factor-rating/). This Threat Factor metric is the number of flights (out of 80, following ABC protocols) in which the birds collide with the treated glass in a forced choice collision trial, where one pane is treated and the other is left untreated. A lower number of treated-pane compared with control-pane collisions indicates less threat to birds. Further, the ABC requires 80 completed collision trials using wild bird that represent any combination of species that were caught nearby to their testing facility. There is not a specific statistical basis for this choice of 80 flights and the sample sizes for individual species are not reported (*Sheppard, 2019*). Hence, we do not know how our sample sizes nor statistical power for our single species (the zebra finch) experimental design compares to ABC methodology, but we assume we have smaller single species samples. Nevertheless, given that we have statistical support for fewer collisions with treated windows for the BirdShades product, we found it useful to convert our data into a metric of Threat Factor to allow for more comparison to other tested window film products. If you accept that there are similarities in our approaches in this part of our study (though smaller sample sizes), conversion of our data to a Threat Factor-like metric results in the following scores: BirdShades = 10 (*i.e.,* 10% of collisions were with the treated window), Haverkamp = 25 (*i.e.,* 25% of collisions were with the treated window). For reference, ABC considers a product whose Threat Factor is below 30 as "bird-friendly".

It is difficult to convert the binary choice collision data to avoidance of collisions with windows. Notably, our binary choice collision trial data suggest a greater reduction in threat than we observed with our collision avoidance metric. A 9-fold (BirdShades) and 3-fold (Haverkamp) reduction in "threat" is much greater than the approximate 39–47% increase in collision avoidance that we quantified even when studying the same birds in the same experimental arena. Admittedly, we are not observing birds in free-living situations: rather, we are attempting to simulate the sensory environment that birds would experience while flying toward a building with windows in the real world—windows are presented in a building structure with daylight on the exterior surface of the window and artificial lighting on the interior surface. Importantly, we allow birds to take avoidance measures rather than forcing collision (as in ABC trials). In either forced-choice or our binary choice trials, birds are likely trying to make a flight decision to make the best of a bad choice where the conspicuous window pane is inflating collisions with the untreated pane. In addition, we studied the flight behaviors of a domesticated species where all individuals were accustomed to captivity and the presence of humans—thereby reducing the influence of captivity and handling stress on flight behavior. In a previous study, we indicated that window collision and avoidance flight behaviors are similar in the domesticated zebra finch to wild-caught brown-headed cowbirds (*Molothrus ater*) (*Swaddle et al., 2020*). Given the direction of the discrepancy between our binary choice collision data and that generated from the collision avoidance trials, we conclude that our computed metrics from binary choice trials overestimate collision reduction and caution that more data from real-world

situations are needed before we can quantify the effectiveness of window treatments in reducing bird collisions.

Considering all of the avoidance and binary choice collision data together, it seems that both the BirdShades and Haverkamp films are likely to lead to increased avoidance and reduced collision with windows; though support for the Haverkamp product is somewhat weaker than the BirdShades film. The Haverkamp film is comprised of lines of colored diamonds—black diamonds next to orange diamonds—and is clearly visible to humans as well as birds. The BirdShades film employs scattering of light in short, ultraviolet wavelengths and appears transparent to human eyes. As most songbirds and several other taxa of birds can see these short wavelengths (*Bennett & Cuthill, 1994*; *Hunt et al., 1998*; *Goldsmith & Butler, 2005*; *Hart & Hunt, 2007*; *Werner et al., 2012*; *Casalía et al., 2021*; *Olsson et al., 2021*), the BirdShades product might be a popular aesthetically-neutral film in many types of installations and especially where songbirds are striking windows during the daytime. Many window collisions occur early in the morning (*Riding, O'Connell & Loss, 2021*). At this time of day the relative proportion of shorter compared with longer wavelength light is at its greatest (*Emerson et al., 2022*), hence UV-affecting products might be particularly conspicuous to birds during this period of greatest daytime collision.

No matter which film is applied to windows, Haverkamp or BirdShades, the films are effective only when applied to the exterior surface of the windows. When the films are applied to the interior surface there is no noticeable avoidance of windows and collision preferences observed in binary choice trials disappear. Therefore, it is essential that end-users of these products install these films on the exterior surface of windows to render any protective benefit. In many situations, it might be more expensive and logistically more difficult to install a window film on an exterior surface, but it is important to remember that installation on an interior surface likely removes the conservation and anti-collision benefits of purchasing and installing the films.

We predicted that internal-surface installation of the Haverkamp product would lead to more avoidance compared with the BirdShades product, but we have no support for that prediction. Even though the Pella replacement windows we used are reported by the manufacturer to reflect a large amount of shorter wavelength light (*Pella, 2022*), it appears that the windows we used disrupt (likely block) light sufficiently across shorter to longer wavelengths that neither film product offers a benefit when installed on the interior surface of the windows. It is also likely that specular glare created from the unaltered external surface of these windows is sufficient that there is some form of a reflected scene visible to birds no matter which product is applied to the internal surface. But when either product is applied to the external surface this reflected scene is disrupted and birds are more likely to avoid collision.

We believe this is the first experimental evidence that supports the recommendation that window films must be installed on exterior glass surfaces to render their collision-avoidance benefits. We expect this finding to be generalizable to other window and film product combinations, as the two film products we tested covered a broad range of avian-visible light spectra (from shorter wavelength UV to mid-range wavelength orange) and many commercially available double-glazed windows are designed to reflect and

transmit light in a similar manner to the windows we used in these experiments. Perhaps only in situations where the birds can see all the way through a transparent structure, such as at a glass-paned bus stop or a glass-sided bridge or walkway, would there be benefits of installing films and products on interior surfaces. Even in these situations there may be filtering of shorter wavelengths of light. Further, in a previous study we showed that window collision and avoidance behaviors are similar between captive zebra finches and wild-caught brown-headed cowbirds (*Swaddle et al., 2020*), hence we predict our results to have some generalizability across species.

## CONCLUSIONS

In conclusion, we provide evidence that the BirdShades and, to a lesser degree, the Haverkamp window film promote avoidance of window collision in a controlled setting. Consumers could likely choose either product and be reasonably confident that they will see a reduction in window collisions, as long as the films are applied to the exterior surface of windows. We conclude that installing window films to interior surface of double-glazed windows will rarely render any collision-reduction benefits and should be avoided in most situations. Further, we advocate for window collision testing protocols that allow for birds to avoid structures, as we found much lower estimates of effectiveness when testing for collision avoidance rather than in situations where collisions are forced. We speculate that many of the published estimates of the effectiveness of window films and decals that used forced collision paradigms render overestimates of how birds will avoid windows in more real-world settings.

## ACKNOWLEDGEMENTS

We thank Shirley Mitchell and Gwendolyn Carter for assistance with animal care.

### Funding
Funding for the test of the BirdShades window film was provided by BirdShades Innovation GmbH. The funders had no role in study design, data collection and analysis, decision to publish, or preparation of the manuscript.

### Grant Disclosures
The following grant information was disclosed by the authors:
BirdShades Innovation GmbH.

### Author Contributions
- John P. Swaddle conceived and designed the experiments, performed the experiments, analyzed the data, prepared figures and/or tables, authored or reviewed drafts of the article, and approved the final draft.
- Blythe Brewster performed the experiments, authored or reviewed drafts of the article, and approved the final draft.

- Maddie Schuyler performed the experiments, authored or reviewed drafts of the article, and approved the final draft.
- Anjie Su performed the experiments, authored or reviewed drafts of the article, and approved the final draft.

## Animal Ethics

The following information was supplied relating to ethical approvals (i.e., approving body and any reference numbers):

The William & Mary Institutional Animal Care and Use Committee approved all procedures in this study.

## Data Availability

The raw data and code are available in the Supplementary Files.

## Supplemental Information

Supplemental information for this article can be found online at http://dx.doi.org/10.7717/peerj.14676#supplemental-information.

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
