# Peer review of "Window films increase avoidance of collisions by birds but only when applied to external compared with internal surfaces of windows"

_PeerJ, doi:10.7717/peerj.14676_

## Round 0.1 · original submission · Minor Revisions

As you can see, all my reviewers like this. In preparing what I'm sure will be the final version, please remember that once accepted, it moves rapidly to going online. We do not copy-edit. What you send is what is published. So, I recommend a very careful read. I use the professional version of Grammarly to check my text. I'm one of that software's most experienced users and one of its most accurate. Do I make mistakes? Of course.

Thanks for submitting this to us.

Reviewer 1 ·

Basic reporting

meets standards

Experimental design

meets standards

Validity of the findings

The findings appear valid. The statitical approach was somewhat cautiious, and used non-parametric tests of choice data. I wonder if the authors considered combining all trials into a mixed model. Would it be possible to do this and calculate probability of collision for all 5 possible window stimuli (blank, Haverkamp in, Haverkamp out, Birdshades in, Birdshades out) while controlling for other factors (bird as random effect, same stimulus both sides or differeing stimuli).
I suspect the take home conclusions would not differ based on the data presented, but this could provide a more statistically sensitive assessment of the effectiveness of the treatements.

I think the raw data file could use more explanation, explaining what each column title indicates and what is on each tab within the excel file.

I think these issues should be straightforward to address.

Additional comments

This study adds some rigorous assessment of the effectiveness of interior vs exteriro application of bird collision deterrents. This is an important and timely study. Current testing methods are mostly proprietary with insufficient statistical validation, so experimental data like these are highly valuable. The conclusions generally support the conclusion that these products need to be on the exterior surface to be effective.

I encourage the others to consider a mixed model approach to the statistical analysis or indicate why it is not appropriate.

Overall the paper is very clearly written.
Minor comments:
1. Could you clarify what day-lit means for the arena? Is the aviary covered by screen or wire mesh? Open to sky? Or is there clear glass? If glass this should be indicated as it would affect the spectra of light in the testing arena.

2. Is there any reason to think the results may vary for other lighting situations (e.g. simulating nighttime collisions with lighting inside the box and darkness outside? Intuitively I think the Haverkamps might be superior in those conditions, whether interior or exterior but that is an empirical question. This may be worth discussing.

3. Fig 1 is very useful and clear. The graphs in Fig 2 and 3 could be better presented with larger labels, bold font as in Fig 1. Filled bars using gray scale rather than orange would be more clear (at least to me).

·

Basic reporting

The Authors did an excellent job providing supporting literature and context. Moreover, they shared their raw data and clearly described those data. I do think that the authors could describe their results more clearly. As one example, the Results section lists the number of collisions (of either control or treatment windows) relative to the total number of trails. However, the corresponding figure and tests compare the number of strikes on control vs treatment windows (not relative to the the total). It would appear that a single bar (BirdShade interior avoidance) in figure 3 is higher than I would expect from the raw data (I expect a value of 0.059 but the bar is >0.1). Moreover, I would recommend that the birds which avoided the windows entirely be considered more explicitly, as the ultimate goal of these window treatments is to increase this subpopulation (i.e., those birds that do not strike buildings). Currently, the comparisons are between the control and treatment for birds that did strike. As a case in point, if a bird flying toward a control window (control on the left, treatment on the right) sees a treatment in their periphery, they may successfully veer toward the left (as flying to the right wouldn't make sense) and avoid striking any surface. While we may not know which window tipped them off, the fact that they avoided the windows is crucially important. Ultimately, I think the contribution would be improved by considering all trials, including those birds that successfully avoided the windows.

Additional details are provided in the attached file.

Experimental design

The experimental design was simple and strong. If possible, it might benefit the paper to explain other forms of controls that were used to prevent conditioning (i.e., using the same birds that may learn to always avoid the wall). Were blank trials interspersed so they could sometimes fly straight? More details in the attachment.

Validity of the findings

While I do have a variety of questions throughout, I think that this (validity of the findings) is the strongest part of this contribution. Their major findings seems strong, well articulated, and very unlikely to change.

Additional comments

Please see the attachment with some minor recommendations.

·

Basic reporting

Writing is clear in general. One comment is to avoid referring to “increase bird collision avoidance” because it is confusing. Instead try “prevention of collisions”/ “decrease in collisions”, or something that doesn’t include conflicting words (increase and avoidance). I understand what you mean but I invite you to explore other ways of saying it.
The last paragraph in the intro seems a bit redundant with the predictions mentioned twice. Merge the sentences and just mention your predictions once.

Experimental design

Experimental design is robust but I was slightly confused when reading about the three different experiments (a, b, c). Not sure why you needed the three experiments and what they contribute. I am certain this can be fixed with a bit of explanation.

Validity of the findings

This article fills a small yet important gap in the study of bird-window collisions and gives us tools to explain why applying the films to the outside is what works. I congratulate the authors on successfully testing this. The paper is sound, succinct, and rigorous and I have only minor comments.
One thing I wonder is if the fact that the zebra finches are captive affects their collision behavior. In another paper we tested if synanthropy (affinity to humans) for some species affected their collision risk. We found that synanthropic species (house sparrows, rock pigeons) though very common, never collide. This is because they are used to humans and the environment. I wonder if this is the case with zebra finches and I invite the authors to further address this is the discussion. They do to some extent now but could be expanded maybe with a reference from Wittig, Thomas W., et al. "Species traits and local abundance affect bird-window collision frequency." Avian Conservation and Ecology 12.1 (2017): 17.

---

## Round 0.2 · accepted · Accept

That you for addressing the reviewers' concerns promptly and in full.